# Long-Term Pulmonary Damage in Surviving Antitoxin-Treated Mice following a Lethal Ricin Intoxication

**DOI:** 10.3390/toxins16020103

**Published:** 2024-02-12

**Authors:** Yoav Gal, Anita Sapoznikov, Shlomi Lazar, David Shoseyov, Moshe Aftalion, Hila Gutman, Yentl Evgy, Rellie Gez, Reinat Nevo, Reut Falach

**Affiliations:** 1Department of Biochemistry and Molecular Genetics, Israel Institute for Biological Research, Ness-Ziona 74100, Israel; anitas@iibr.gov.il (A.S.); moshea@iibr.gov.il (M.A.); yentle@iibr.gov.il (Y.E.); 2Department of Pharmacology, Israel Institute for Biological Research, Ness-Ziona 74100, Israel; shlomil@iibr.gov.il (S.L.); hilag@iibr.gov.il (H.G.); rellia@iibr.gov.il (R.G.); 3Pediatric Pulmonology Unit, Hadassah Medical Center, Jerusalem P.O. Box 12000, Israel; dshoseyov@gmail.com; 4Department of Biomolecular Sciences, Weizmann Institute of Science, Herzel 234, Rehovot P.O. Box 26, Israel; reinat.nevo@weizmann.ac.il

**Keywords:** ricin, intranasal, antitoxin, long-term damage

## Abstract

Ricin, a highly potent plant-derived toxin, is considered a potential bioterrorism weapon due to its pronounced toxicity, high availability, and ease of preparation. Acute damage following pulmonary ricinosis is characterized by local cytokine storm, massive neutrophil infiltration, and edema formation, resulting in respiratory insufficiency and death. A designated equine polyclonal antibody-based (antitoxin) treatment was developed in our laboratory and proved efficacious in alleviating lung injury and increasing survival rates. Although short-term pathogenesis was thoroughly characterized in antitoxin-treated mice, the long-term damage in surviving mice was never determined. In this study, long-term consequences of ricin intoxication were evaluated 30 days post-exposure in mice that survived antitoxin treatment. Significant pulmonary sequelae were demonstrated in surviving antitoxin-treated mice, as reflected by prominent histopathological changes, moderate fibrosis, increased lung hyperpermeability, and decreased lung compliance. The presented data highlight, for the first time to our knowledge, the possibility of long-term damage development in mice that survived lethal-dose pulmonary exposure to ricin due to antitoxin treatment.

## 1. Introduction

Ricin is a type II ribosome-inactivating protein (RIP) obtained from the seeds of *Ricinus communis* (castor beans). The complete toxin comprises two polypeptide chains (A and B) connected by a disulfide bond. The B chain is a lectin that attaches to galactose residues on cell surfaces. The A chain possesses RNA N-glycosidase activity, leading to the irreversible inactivation of the 28S rRNA of the mammalian 60S ribosome sub-unit, thereby halting cell protein synthesis [1]. Due to its widespread availability and the relatively simple production process, ricin is considered a potential biological threat agent [2].

The severity of ricin toxicity varies depending on the route of exposure, with inhalation being the most serious [3]. Pathological studies of pulmonary ricin intoxication have shown that the damage is mainly localized to the lungs [4]. This is characterized by a local cytokine storm, extensive neutrophil recruitment, increased pro-oxidant enzyme activity, and the development of proteinaceous pulmonary edema, ultimately leading to respiratory failure and death. These clinical features are associated with acute lung injury [5], which can progress to acute respiratory distress syndrome (ARDS), as observed in swine [6].

Previous research has explored the potential of protecting mice from pulmonary ricinosis via passive immunization using polyclonal anti-ricin antibodies [5,7]. In the past few years, we have produced F(ab’)_2_-based ricin antitoxin made from the hyperimmune plasma of a horse that was immunized with inactivated (monomerized) ricin. We demonstrated that treating mice with this antitoxin preparation results in significantly enhanced survival. The treatment based on equine antitoxin has demonstrated effective anti-edematous and anti-inflammatory effects, apparently by balancing cytokine levels in the lungs of intoxicated animals [8]. While mice survival following pulmonary exposure to ricin and antitoxin administration were extensively studied, displaying different survival rates depending on the treatment time [9], currently, there are no data regarding the long-term sequelae in the surviving mice. In a scenario where an antitoxin treatment prevents mortality, it is important to know whether functional chronic lesions develop in the lungs over time. If such damage indeed occurs, a thorough characterization of the underlying long-term pathogenesis may be highly valuable for developing proper therapeutic countermeasures.

Long-term damage was reported in acute lung injury models in mice, i.e., pulmonary exposure to LPS. Persistent lung inflammation increased total lung volumes, and the presence of lung remodeling with collagen deposition as indicative of fibrotic process was determined five weeks following exposure to nebulized endotoxin [10]. In human beings, physical, functional and even cognitive long-term damages (five years) were reported following ARDS. In particular, patients with more severe ARDS had significantly lower pulmonary function scores one year following ARDS in comparison to other survivors throughout the follow-up period [11,12].

In this respect, we found significant levels of damage markers in the lungs of antitoxin-treated mice even at 72 h following intoxication [8]. Thus, the clinical manifestation itself may create irreversible damage to the lungs, with long-term sequelae even after neutralizing the toxin by the antitoxin treatment.

The goal of this study was to characterize the long-term damage in mice that survived intranasal ricin intoxication following antitoxin treatment.

## 2. Results

To characterize the long-term pathophysiological consequences following intranasal ricin intoxication, we determined gross physiological changes (body weight and temperature), as well as parameters of pulmonary damage. Mice were intranasally challenged with a lethal dose of ricin, treated 24 h post-exposure (PE) with anti-ricin antitoxin, and monitored for 30 days. Long-term analysis was performed on surviving animals.

### 2.1. Long-Term Body Weight and Body Temperature Changes in Antitoxin-Treated Mice That Survived Ricin Intoxication

The body weight (BW) of surviving mice was monitored every day or every other day for 30 days following intranasal ricin intoxication. Antitoxin treatment was administered at 24 h PE. As seen (Figure 1A), a decrease of more than 25% in BW was recorded a week following intoxication in antitoxin-treated in comparison to healthy mice. The reduction in average weight of the antitoxin-treated group was similar to that recorded in intoxicated mice that were not treated (up to the time points of death for ricin-exposed, non-treated mice). The average BW of the antitoxin-treated group gradually returned to normal values, similar to those of healthy animals, about four weeks following intoxication.

The body temperature of the antitoxin-treated mice decreased for about a week following intoxication, similar to BW and then returned to normal values within another week (Figure 1B). Ricin-intoxicated mice that were not treated with antitoxin succumbed during the first week following intoxication (Figure 1A,B).

### 2.2. Pulmonary Damage Characterization in Surviving (Antitoxin-Treated) Mice

#### 2.2.1. Histological and Immunohistological Evaluation

Histological inspection of the lungs of surviving mice treated with antitoxin was conducted at day 30 PE in comparison to the proper structure of lungs in healthy mice (Figure 2A,B). As exemplified by hematoxylin and eosin (H&E) staining, profound pathological changes were detected in the lungs of ricin-intoxicated mice. We observed prominent focal damage in the lungs of ricin-intoxicated and antitoxin-treated mice, including intensive cell infiltration, alveolar epithelial disruption, edema, platelet aggregates in small pulmonary blood vessels and capillaries, and occasional inflammatory cellular depositions (Figure 2C,D). Accordingly, the total histology severity score was quantified by analysis of parameters such as the presence of foam cells, fibrinous deposits, hemorrhages in air compartments, vessel congestion, alveolar wall thickness, and leukocyte infiltration. This abnormal score in the lungs of surviving mice indicates severe long-term damage (Figure 2E).

In the next step, we analyzed the fibrotic state of the lungs 30 days post-ricin exposure and treatment using Masson’s Trichrome (MT) staining. In ricin-exposed antitoxin-treated mice, collagen accumulation in the lungs was evident. Fibrotic foci were detected mostly in the enlarged perivascular and peribronchial areas (Figure 3C,D), in comparison to intact non-fibrotic lung tissue of healthy mice (Figure 3A,B). Overall, quantification of fibrotic foci showed that the percentage of the fibrotic areas in the lungs of ricin-intoxicated and antitoxin-treated mice was elevated (Figure 3E).

Next, we examined the long-term effect on parenchymal cell populations in the lungs of ricin-exposed antitoxin-treated mice. Within the parenchymal cells, we distinguished between vascular endothelial cells and alveolar epithelial cells. In the endothelial (data not shown) and alveolar epithelial type I cell population (Figure 4A,B), we did not observe any long-term damage at 30 days PE. However, a significant increase in alveolar epithelial type II cell numbers was observed 30 days PE following antitoxin treatment, in comparison to sham mice (Figure 4C–E).

#### 2.2.2. Lung Weight to Body Weight Ratio

To further characterize long-term damage in surviving mice that were treated with antitoxin, the lungs and mice were weighed, and the lung wet weight/body weight ratio (LWW/BW) was calculated at 30 days PE. This ratio reflects the degree of injury, i.e., edema. As can be seen (Figure 5), LWW/BW was significantly elevated in ricin-intoxicated surviving mice 30 days PE (923 ± 146 mg/g) in comparison to the sham group’s LWW/BW (604 ± 23 mg/g), suggesting tissue hyperpermeability and edema. This finding further supports the notion of long-term pulmonary damage in ricin-intoxicated antitoxin-treated mice that survived pulmonary ricin exposure.

#### 2.2.3. Lung Compliance

In order to examine the functionality of lungs in convalescing mice, we determined lung compliance 30 days PE. Lung compliance is a functional pulmonary measure correlating to the ability of the lungs to stretch and expand as quantified by a specialized device designed for this purpose. Low compliance indicates stiffer lungs, such as in the case of pulmonary fibrosis [13,14,15,16]. As can be seen (Figure 6), lung compliance decreased significantly in the antitoxin group 30 days PE, in comparison to the sham group. We previously demonstrated that the co-administration of ciprofloxacin with antitoxin significantly alleviated inflammation and edema formation, as well as survival rates of intoxicated mice [17]. Thus, we tested whether concomitant antitoxin–ciprofloxacin treatment would improve lung compliance 30 days PE. As seen in Figure 6, the ciprofloxacin–antitoxin combination significantly improved lung compliance in comparison to antitoxin treatment alone. This combined treatment led to full functional restoration of the lung, in which we found no significant difference between lung compliance of the ciprofloxacin–antitoxin group to the sham group.

## 3. Discussion

Previously, we and others have thoroughly investigated the short-term pathophysiological process following pulmonary ricin intoxication (3–4 days PE). This is characterized by cytokine storm, massive neutrophil recruitment, increased oxidative and proteolytic damages, and development of proteinaceous edema (determined as early as 24 h PE), all of which are spatially restricted to the lungs and associated with respiratory failure and death [5,6]. In contrast to short-term damage, no studies were conducted to characterize long-term damage in lethally intranasally intoxicated mice who survived intoxication following antitoxin treatment.

It should be mentioned that although 30 days PE is considered a long period of time in mice [18,19], the damage (in particular pulmonary damage) could worsen at later stages, i.e., 60 days PE or later [20]. Vice versa, it could be that following a longer period of time, enhanced tissue repair leads to decreased damage.

Studies have shown that preexisting antibodies against ricin, even at relatively high doses in the BAL fluids, were not sufficient to fully suppress toxin-induced inflammation and pulmonary damage associated with ricin exposure. Additionally, high doses of monoclonal anti-ricin antibody treatment did not fully negate morbidity associated with toxin insult, even when mice invariably survived intoxication [21,22]. Moreover, monoclonal antibody treatment of non-human primates following lethal dose aerosol ricin challenge revealed evidence of chronic inflammation and distinctive fibrosis proximal to the respiratory bronchioles 21 days PE [23,24].

In this study, the antitoxin treatment was given at late times following pulmonary exposure (24 h PE). Accordingly, the therapeutic intervention was administered in parallel to the development of intense pulmonary inflammation. Thus, it was probable that ricin neutralization per se would not be sufficient for complete lung damage resolution and that irreversible deterioration would lead to long-term damage. In this respect, it should be mentioned that no evidence was observed for long-term damage determined 14 days PE in animals immunized with recombinant ricin A sub-unit vaccine (RiVax) following exposure to high concentrations of aerosolized ricin [25], where prompt toxin neutralization (prior to damage propagation) is expected to result in attenuated inflammatory damage in comparison to unvaccinated animals.

Indeed, in the present study, we demonstrated, for the first time, that quantifiable long-term damage is formed in mice intranasally exposed to a lethal dose of ricin and successfully treated with antitoxin administered at relevant yet late time points post-exposure. This damage was characterized by histopathological and functional impairments of the lungs. The histopathological evaluation provided evidence of prominent focal damage in the lungs, including inflammation, alveolar epithelial disruption, and edema that altogether resulted in an abnormal histological score. In addition, moderate pulmonary fibrotic foci formation was detected. These findings indicate severe long-term damage in the lungs of surviving mice. It should be mentioned that this damage is vastly different from short-term damage (i.e., the damage recorded at 72 h PE) since there are no neutrophils (Figure 2), pro-inflammatory cytokines, nor oxidative and proteolytic damage markers detected at 30 days PE (data not shown). Immunohistochemical differential examination of parenchymal cell populations at 30 days PE demonstrated hyperplasia of alveolar type II epithelial cells, which could be indicative of lung repair and recovery. In this regard, alveolar type II epithelial cells are one of the major targets of ricin-induced depurination [26,27]. These cells have an important function in pulmonary regeneration as well as local defense mechanisms of the respiratory system. In addition, alveolar type II epithelial cells produce cytokines and growth factors that could affect immune cells and prevent edema via active sodium transport in sodium channels [28]. Furthermore, ineffective repair of damaged alveolar epithelium, in particular type II epithelial cells, has been postulated to cause pulmonary fibrosis in mice [29]. Accordingly, increased numbers of type II epithelial cells in the antitoxin-treated mice 30 days PE may represent enhanced attempts of tissue repair and remodeling, which occurred in parallel to the fibrotic processes.

Lung compliance is a functional respiratory parameter, which was never evaluated, to the best of our knowledge, following ricin intoxication in mice. In this work, we designed and employed a designated apparatus for lung compliance measurement. We demonstrated that lung compliance of ricin-intoxicated mice saved by antitoxin treatment significantly decreased, supporting the presence of long-term lung damage. The concomitant administration of ciprofloxacin with the antitoxin, a combination that was previously demonstrated to improve both survival and acute lung injury-associated pathology [17], significantly protected lungs from long-term damage, as the compliance of ciprofloxacin–antitoxin-treated mice was comparable to that of sham animals.

In summary, long-term functional and histopathological damages were observed in ricin-intoxicated mice successfully treated with antitoxin. The concomitant treatment of immune modulating- or anti-fibrotic drugs (such as pirfenidone and nintedanib) [30], should be considered as a medical countermeasure for long-term damage prevention.

## 4. Materials and Methods

### 4.1. Ricin Preparation

Crude ricin was prepared as described before [31]. Briefly, to obtain crude ricin, seeds of the endemic *Ricinus communis* were homogenized in a Waring blender (Waring, Torrington, CT, USA) with 5% acetic acid (Merck, Darmstadt, Germany) in PBS (Biological Industries, Beth-Haemek, Israel). The homogenate was then centrifuged, and the clarified supernatant containing the toxin was precipitated with 60% saturated ammonium sulfate (Merck, Darmstadt, Germany). The precipitate was then dissolved in PBS and extensively dialyzed against the same buffer. The toxin appeared on a Coomassie blue (Bio-Rad, Rishon Le Zion, Israel)-stained non-reducing 10% polyacrylamide gel (Thermo Fisher Scientific, Carlsbad, CA, USA) as two major bands with molecular weights of approximately 65 kDa (ricin toxin, ~80%) and 120 kDa (*Ricinus communis* agglutinin, RCA, ~20%). The protein concentration was determined to be 2.86 mg/mL by 280 nm absorption using a Nanodrop (Nanodrop 2000; Thermo Fisher Scientific, Waltham, MA, USA). 

### 4.2. Anti-Ricin Antitoxin

The Fc regions of pooled horse hyperimmune antisera were cleaved with pepsin (1200 U/mL, pepsin A from porcine stomach mucosa, Sigma, Steinhaim, Germany) at 30 °C and pH 3.2, and then at 18 °C and pH 7.4. Purification of the F(ab’)2 fragments was conducted as follows: The contaminating proteins were precipitated with ammonium sulfate (25% saturation for 20 h at 18 °C). Then, the contaminating proteins were separated from the suspension, and the filtrate of the microfiltration was washed (from small contaminating proteins and peptides), concentrated, and dialyzed (against 50 mM phosphate buffer pH 8). The dialyzed solution was applied on a Q-sepharose anion-exchange column (Q sepharose fast flow, GE Healthcare, Uppsala, Sweden) so that the F(ab’)2 fragments fraction was eluted with the flow-through and collected. The F(ab’)2-containing flow-through solution was adjusted to pH 6 and applied onto a SP-sepharose cation-exchange column (SP sepharose fast flow, GE healthcare, Uppsala, Sweden). The F(ab’)2-containing flow-through was collected. All chromatographic processes were carried out using an AKTA Process Instrument (GE Healthcare, Uppsala, Sweden). Finally, the antitoxin solution was concentrated, dialyzed (against 300 mM glycine buffer, pH 7.4), and filtered using a nano filter (Kleenpak-Nova, PALL Life Science, New York, NY, USA) [8].

### 4.3. Animal Studies

The experiments involving animals were conducted in compliance with Israeli law and were approved by the Institutional Animal Care and Use Committee (IACUC) of the Israel Institute for Biological Research (Ness-Ziona, Israel) under the following protocol numbers: M-27-16, M-77-16, M-83-16, M-78-17. The treatment of the animals adhered to the regulations outlined in the USDA Animal Welfare Act and the conditions specified in the National Institute of Health Guide for Care and Use of Laboratory Animals. All the animals used in the study were female CD-1 mice (Charles River Laboratories Ltd., Margate, UK) weighing 27–32 g. They were housed in filter-top cages in a controlled environment, maintained at 21 ± 2 °C and 55 ± 10% humidity, with lighting set to simulate a 12/12 h dawn to dusk cycle. The mice were acclimatized in a suitable animal holding facility for 4–8 days before the start of the experiment and had access to food and water ad libitum. For intoxication, mice were anesthetized via an intraperitoneal (i.p.) injection of ketamine (1.9 mg/mouse) and xylazine (0.19 mg/mouse). Subsequently, crude ricin (50 µL; 7 µg/kg diluted in PBS) was administered intranasally (i.n., 2 × 25 µL), and the mortality of the mice was observed over a 30-day period. Prior to these studies, we determined that 3.5 µg crude ricin/kg body weight is approximately equivalent to 1 mouse (intranasal) LD_50_ (95% confidence intervals of 2.3 to 4.5 µg/kg body weight).

Antibody administration was performed on mice anesthetized as above. For antibody treatment, a volume of 50 µL of anti-ricin antibody preparation was delivered i.n. (2 × 25 µL) at 24 h following intoxication. The sham control are naïve mice that were administered PBS (2 × 25 µL) at day 0 and 24 h later. Ciprofloxacin (Ciprofloxacin-Teva, 200 mg/100 mL, Or-Akiva, Israel) was administered at a dose of 200 mg/kg body weight (3 mL, i.p.) at 24, 48, 72, and 96 h post-exposure.

The determination of lung compliance (Figure 6) was conducted as a separate experiment.

### 4.4. Histology

Lungs perfused with PBS were harvested and immersed in neutral buffered formalin (4%, 2 weeks, room temperature) prior to paraffin embedment. Then, lung sections (5 µm) were mounted on glass slides and stained with hematoxylin and eosin (H&E). The lung histology severity score was based on an assessment of the following parameters: the presence of foam cells, fibrinous deposits, hemorrhages in air compartments, vessel congestion, alveolar wall thickness, and leukocyte infiltration. Ten images of at least five random areas were taken under the same magnification (×20). Each parameter was graded [0–4: 0 for normal, 4 for severe damage]. The total histology severity score was calculated by summing all parameters’ grades (0–24; a higher score is indicative of more severe damage). For fibrosis visualization, Masson’s Trichrome (Sigma-Aldrich-Merck, Jerusalem, Israel) staining was performed according to the manufacturer’s instructions. The fibrosis percentage was calculated using Fiji software (version 2.15.0, Rasband, W.S., ImageJ, U.S. National Institutes of Health, Bethesda, MD, USA, 2017). The stack command was used to split the image and apply the threshold and montage commands to view all three channels at the same time. We chose to continue the analysis with the green channel, which had the best separation. Since Fiji is not able to correctly threshold this image, the threshold was manually adjusted (0–62), and the percent fibrotic area was measured.

### 4.5. Immunohistochemistry

Lungs perfused with PBS were collected and placed in neutral buffered formalin (4%, 2 weeks, room temperature) before being embedded. Paraffin sections (5 µM) were mounted on glass slides and deparaffinized via a gradient of 100%, 95%, 70%, and 30% ethanol, washed in distilled water, and the antigens were retrieved using a commercial antigen retrieval solution (Dako, Carpinteria, CA, USA). Sections were permeabilized for 10 min (0.2% Triton X-100 in PBS), blocked for 1 h (10% normal goat serum in PBS containing 0.05% Triton X-100), and incubated in purified rabbit polyclonal anti-pro-SPC primary antibody (Merck, Rehovot, Israel) or purified anti-podoplanin (T1alpha, 8.1.1, eBioscience, San Diego, CA, USA) in an antibody cocktail solution (50% blocking solution, 0.05% Triton X-100 in PBS) for 24 h at 4 °C. Sections were then washed with washing buffer three times (1% blocking solution, 0.05% Triton X-100 in PBS) and incubated with anti-rabbit Alexa Fluor 594 secondary antibody (Molecular Probes, Burlington, NJ, USA) or with FITC-coupled goat anti-Armenian hamster antibody (Molecular probes, Burlington, NJ, USA) in antibody cocktail solution for 1 h at RT. Nuclei were stained with DAPI, and the slides were washed three times. Next, slides were mounted using Fluoromount G (Southern Biotech, Birmingham, AL, USA), and images were acquired using a Zeiss LSM710 confocal microscope (Zeiss, Obercochen, Germany). Image analysis was performed using QuPath (version 0.4.0, Edinburgh, UK, 2022) [32]. Nuclei were segmented using Cellpose (version 2.0, Ashburn, VA, USA, 2022) [33] with a cyto-pretrained model for the DAPI channel, and the cellular border was extrapolated at a distance of 5 pixels from the nuclear border. The number of positive pro-SPC cells was calculated by object classification using a random tree.

### 4.6. Assessment of Lung Compliance

Thirty days PE, surviving antitoxin-treated or Sham-treated mice were euthanized, tracheotomized, and a cannula was inserted into the trachea above the bifurcation. A homemade device for lung compliance measurement was designed: two plastic tubes were inserted into a T-shaped connector, one of which was connected to the cannula, and the second to a monometer (a U-shape tube filled with colored liquid). A plastic syringe was connected to the T-splitter, and 1 mL of air was injected into this apparatus. The calculation of compliance was conducted as follows: compliance = ∆V/∆P, where ∆V is the volume of air injected into the T-shaped connector (1 mL), and ∆P is the change in pressure (as reflected by the increase in water meniscus of the monometer). The shift (rise) in the liquid meniscus was inversely correlated to lung compliance.

### 4.7. Statistical Analysis

Statistical analyses were conducted using GraphPad Prism software (version 5.01, GraphPad Software Inc., La Jolla, CA, USA, 2007). Data are represented as means ± SEMs. Significance for comparisons was assessed by unpaired *t*-test analysis or one-way analysis of variance (ANOVA) followed by Tukey’s multiple comparisons test. Differences were considered significant at *p* < 0.05.

## Figures and Tables

**Figure 1 toxins-16-00103-f001:**
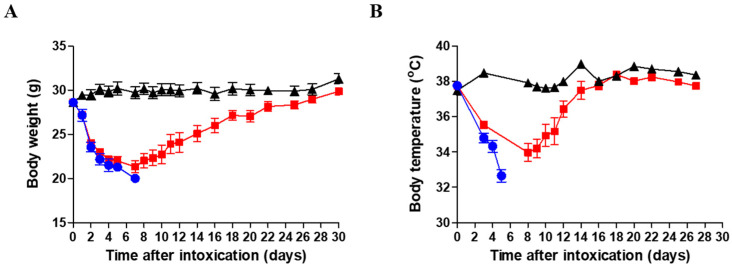
Body weight and temperature of surviving ricin-intoxicated antitoxin-treated mice. Mice were intranasally intoxicated with 2LD_50_ (7 µg/kg) ricin. Antitoxin was administered to indicated groups of mice at 24 h PE. Sham group was administered PBS instead of either ricin or antitoxin. (**A**) Body weight and (**B**) body temperature were monitored for 30 days following intoxication. Sham—black triangles (*n* = 8); ricin—blue circles (*n* = 7); ricin + antitoxin—red squares (*n* = 12–16).

**Figure 2 toxins-16-00103-f002:**
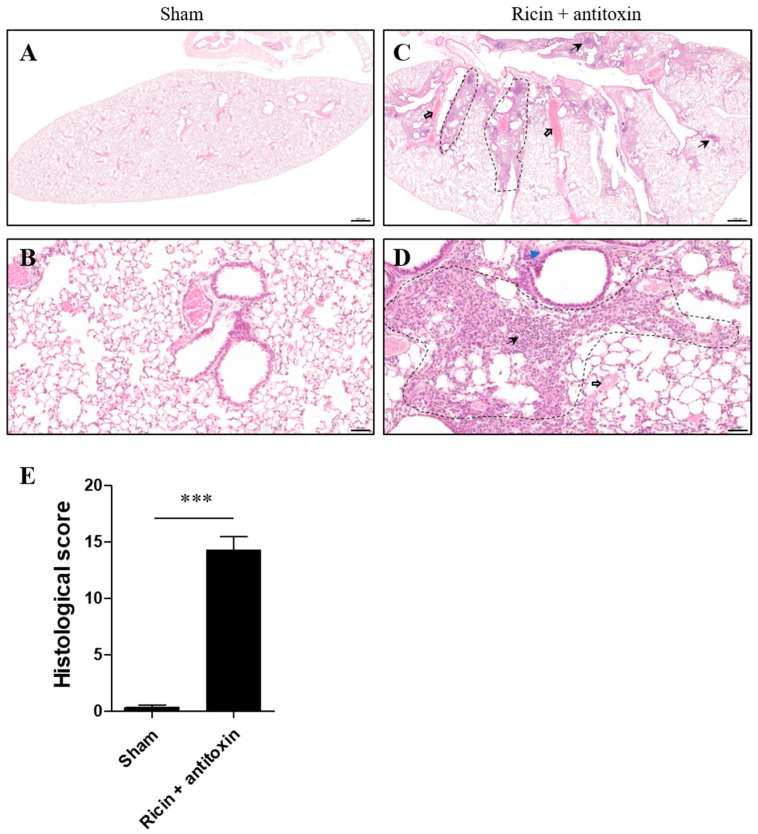
Histological evaluation of lungs of surviving intranasally ricin-exposed and antitoxin-treated mice. Mice were intranasally intoxicated with 2LD_50_ (7 µg/kg) ricin. Antitoxin was administered at 24 h PE. Sham group was administered PBS instead of either ricin or antitoxin. At 30 days PE, lungs were harvested, and histological staining was conducted. Lung H&E staining of (**A**,**B**) sham mice and (**C**,**D**) ricin + antitoxin-treated mice. (**A**,**C**) Magnification ×2; (**B**,**D**) magnification ×20. Scale bar 500 (**A**,**C**) and 50 (**B**,**D**) µm, *n* = 3 per group. Dashed areas represent focal damage: intensive cell infiltration and alveolar epithelial disruption; black thick and thin arrows show platelet aggregates and occasional inflammatory cellular depositions, respectively; blue arrows show edema. (**E**) Histological severity score. Data are expressed as means ± SEM. *** *p* < 0.001.

**Figure 3 toxins-16-00103-f003:**
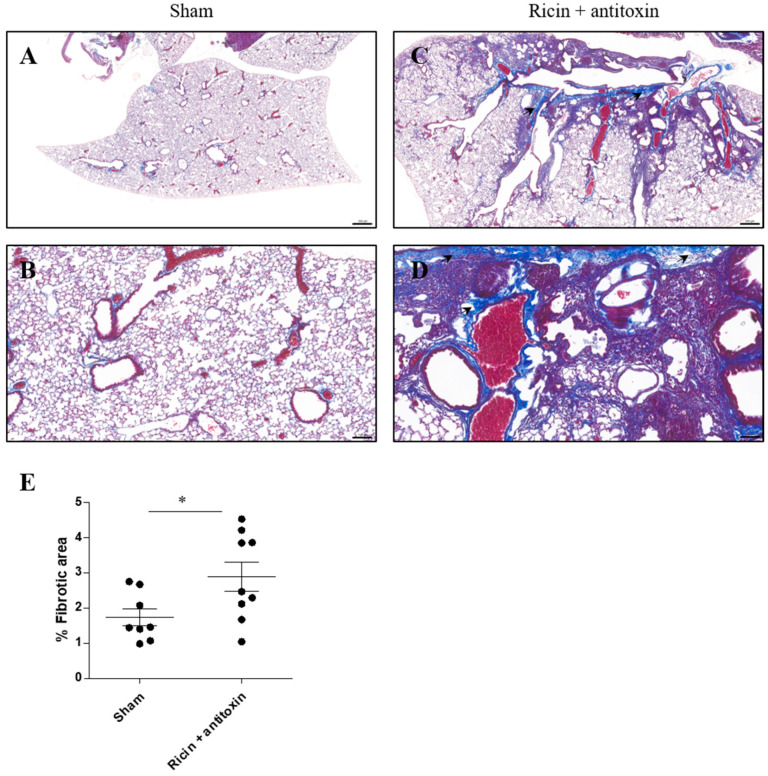
Collagen deposition in the lungs of surviving intranasally ricin-exposed and antitoxin-treated mice. Mice were intranasally intoxicated with 2LD_50_ (7 µg/kg) ricin. Antitoxin was administered at 24 h PE. Sham group was administered PBS instead of either ricin or antitoxin. At 30 days PE, lungs were harvested, and histological staining was conducted. Masson’s Trichrome staining of lung sections in (**A**,**B**) sham and (**C**,**D**) ricin + antitoxin-treated mice. (**A**,**C**) Magnification ×2; (**B**,**D**) magnification ×10. Black arrows represent collagen accumulation and enlarged perivascular and peribronchial areas. (**E**) Percentage of fibrotic areas in lungs of individual mice. Scale bar 500 (**A**,**C**) and 100 (B,D) µm, *n* = 3 per group, 3 fields of view per mouse. Data are expressed as means ± SEM. * *p* < 0.05.

**Figure 4 toxins-16-00103-f004:**
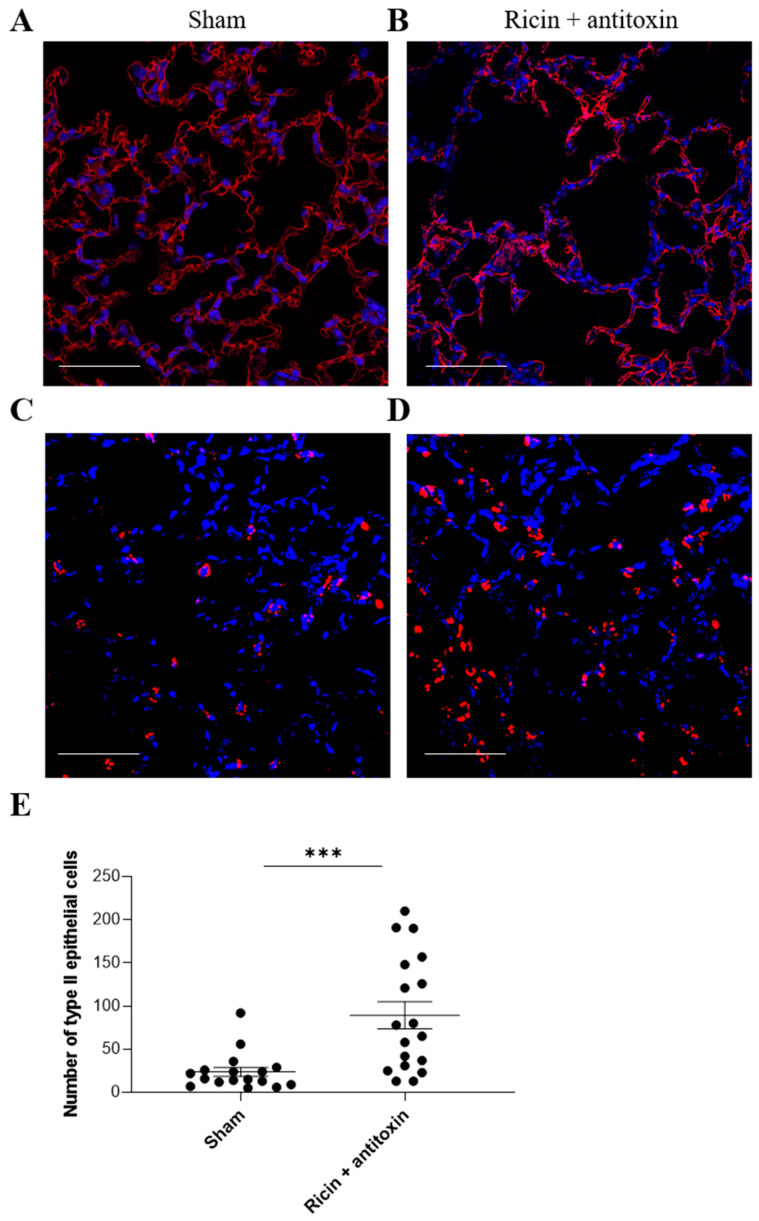
Immunofluorescent analysis of alveolar type I and II cells in the lungs of intranasally ricin-exposed and antitoxin-treated mice 30 days following ricin intoxication. Mice were intranasally intoxicated with 2LD_50_ (7 µg/kg) ricin. Antitoxin was administered at 24 h PE. Sham group was administered PBS instead of either ricin or antitoxin. At 30 days PE, lungs were harvested, and immunohistological staining was conducted for (**A**,**B**) type I epithelial cells (T1α staining in red; DAPI in blue), (**C**,**D**) type II epithelial cells (pro-SPC staining in red; DAPI in blue). (**A**,**C**) Sham and (**B**,**D**) ricin + antitoxin-treated mice. Scale bar 50 µm, *n* = 3 per group. (**E**) Number of pro-SPC^+^ cells per field of view (*n* = 3 per group; 5–6 fields of view per mouse). Data are expressed as means ± SEM. *** *p* < 0.001.

**Figure 5 toxins-16-00103-f005:**
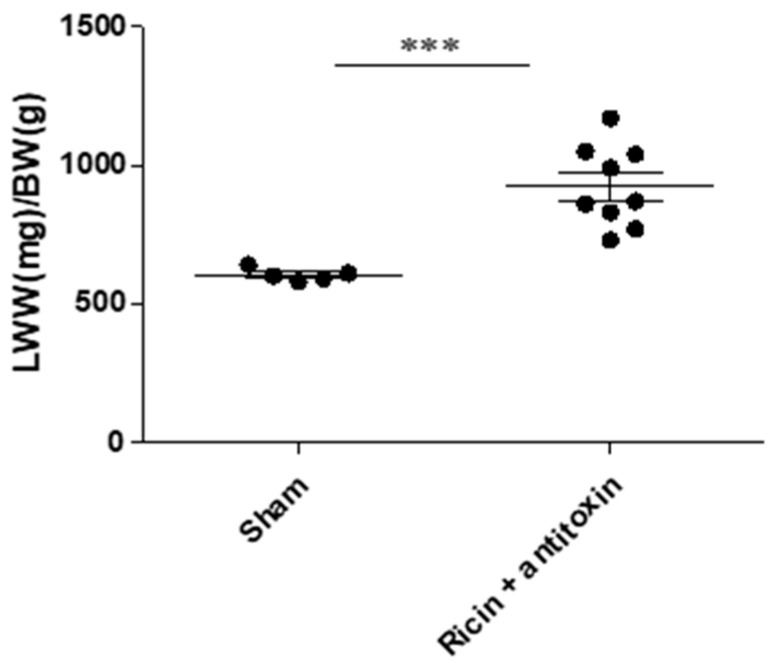
Lung weight to body weight ratio in intranasally ricin-exposed and antitoxin-treated mice 30 days following ricin intoxication. Mice were intranasally intoxicated with 2LD_50_ (7 µg/kg) ricin. Antitoxin was administered at 24 h PE. Sham group was administered PBS instead of either ricin or antitoxin. At 30 days PE, mice were weighed, and following euthanasia, lungs were harvested and immediately weighed. LWW/BW was calculated for each mouse; *n* = 5, 9 per group. Data are expressed as means ± SEM. *** *p* < 0.001.

**Figure 6 toxins-16-00103-f006:**
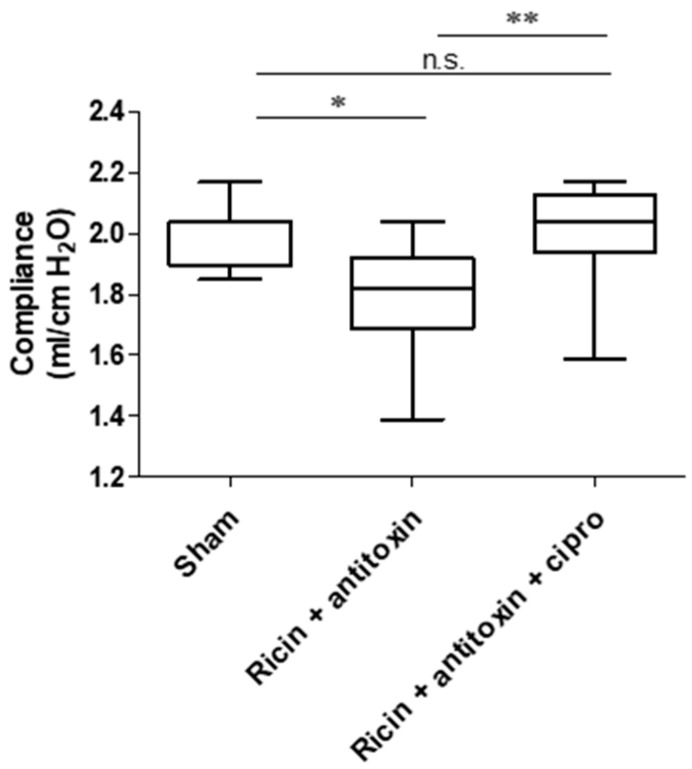
Lung compliance in antitoxin and antitoxin-ciprofloxacin treated mice 30 days following ricin intoxication. Mice were intranasally intoxicated with 2LD_50_ (7 µg/kg) ricin. Antitoxin was administered at 24 h PE. Ciprofloxacin (200 mg/kg) was administered daily to the relevant group intraperitoneally at 24, 48, 72, and 96 h PE. Sham group was administered PBS instead of either ricin or antitoxin. At 30 days PE, lung compliance was determined. *n* = 8, 11, and 12 per group. Data are expressed as means ± SEM. * *p* < 0.05, ** *p* < 0.01, n.s., not significant.

## Data Availability

Data are contained within the article.

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
