# Peer review of "Long-Term Pulmonary Damage in Surviving Antitoxin-Treated Mice following a Lethal Ricin Intoxication"

_toxins, 2024, doi:10.3390/toxins16020103_

Round 1

Reviewer 1 Report

Comments and Suggestions for Authors

1. I think the paper could be accepted as submitted although it should probably be improved in terms of presentation, clarity and transparency (eg not clear if this is now an industry sponsored study; even though detailed methods available in 2018...why make the reader work that hard--just add a sentence or two)...this is an Editor's Decision. 

A. Figure 1: Body weight and temperature: Using different colors to go with the shapes (triangles/squares) will bring contrast to the figure making it easier to follow/disambiguate the lines...seeing that this was done in Prism, the line colors can be changed in seconds (e.g. Black for sham, red for ricin, blue for ricin + antitoxin 

B. Histopath and Functional outcome I am slightly confused why histopath isn't shown for the ricin only controls. They survived six days. This seems like a conspicuous omission in histopath and in scoring. Given that the clinical picture of the animals in terms of functional outcomes at 30 days is good...any clinical result supersedes the histopath/scoring even if differences are hard to detect. As well, there is a temporal difference in times of tissue collection, so again--differences can simply be noted in the limitations that the collection wasn't windowed. I think this raises questions about omission of data. At the very least, this should be addressed in the discussion. It seems unlikely that it was not done as part of the original study plan and that this work would have been done at a facility with high BSL (not mentioned...but probably good for people to realize you cannot just do these types of studies anywhere).

C. Even if the short-term histopath/function was previously studied, showing that it was reproducible/not reproducible/different under different conditions is important information for investigators

3.  Great study. Looking forward to seeing it published. It can be accepted as is or minor improvements that would make it easier to understand/more credible (e.g. ricin histopath/functional scores or explicit statements "why not those?"

4. Low level question: Besides cipro, what other interventions would the authors consider to be of interest? Are there any non-antibiotic adjuncts? 

Author Response

Manuscript ID: toxins 2797472

Response to reviewer 1:

  1. I think the paper could be accepted as submitted although it should probably be improved in terms of presentation, clarity and transparency (eg not clear if this is now an industry sponsored study; even though detailed methods available in 2018...why make the reader work that hard--just add a sentence or two)...this is an Editor's Decision. 

We agree with Reviewer 1. Section 4.2 is described in detail now (we elaborated on the procedure of antitoxin preparation). As for your comment regarding the sponsorship, the antitoxin is used for preclinical research only and is produced locally at the IIBR.

  1. Figure 1: Body weight and temperature: Using different colors to go with the shapes (triangles/squares) will bring contrast to the figure making it easier to follow/disambiguate the lines...seeing that this was done in Prism, the line colors can be changed in seconds (e.g. Black for sham, red for ricin, blue for ricin + antitoxin. 

Figure 1 has been changed and incorporated into the text according to the Reviewer suggestion.

  1. Histopath and Functional outcome I am slightly confused why histopath isn't shown for the ricin only controls. They survived six days. This seems like a conspicuous omission in histopath and in scoring. Given that the clinical picture of the animals in terms of functional outcomes at 30 days is good...any clinical result supersedes the histopath/scoring even if differences are hard to detect.

The goal of this work was to characterize only the long-term damage, thus short-term pathology was not conducted. It should be mentioned, that in contrast to few days following ricin exposure, at 30 days PE one cannot detect neutrophils/perivascular edema, and likewise there were no pro-inflammatory cytokines detected. However, we agree with the reviewer that it is important to discuss this issue, thus we added few words in the “Discussion section” regarding the absent of damage markers at 30 days PE (Lines: 245-248): “It should be mentioned that this damage is vastly different from short-term damage (i.e. the damage recorded at 72 hours PE), since there are no neutrophils (Figure 2), pro-inflammatory cytokines, nor oxidative damage markers detected at 30 days PE (data not shown).”  

As well, there is a temporal difference in times of tissue collection, so again--differences can simply be noted in the limitations that the collection wasn't windowed. I think this raises questions about omission of data. At the very least, this should be addressed in the discussion. It seems unlikely that it was not done as part of the original study plan and that this work would have been done at a facility with high BSL (not mentioned...but probably good for people to realize you cannot just do these types of studies anywhere).

We thank the Reviewer for this excellent remark. All tissues and data (Figures 1-5) were collected simultaneously. However, compliance determination in the presence of ciprofloxacin was conducted as a separate experiment (and we just added previous data of “sham” and “antitoxin”- treated mice). According to your insight, the data now presented in Figure 6 is only from this separate experiment, and we added to section 4.3 (animal experiments) the following sentence: “The determination of lung compliance (Figure 6) was conducted as a separate experiment”. 

Regarding the comment about the high BSL, the work was performed under BSL2 conditions (intranasal intoxication of anesthetized mice).

  1. Even if the short-term histopath/function was previously studied, showing that it was reproducible/not reproducible/different under different conditions is important information for investigators.

We are not sure what the Reviewer means. The short term is highly reproducible as previously shown by us and others. The experimental conditions for short-term or long-term damage determination are very similar regarding the toxin (same LD50), the antitoxin (same batch and treatment mode) and so on.

  1.  Great study. Looking forward to seeing it published. It can be accepted as is or minor improvements that would make it easier to understand/more credible (e.g. ricin histopath/functional scores or explicit statements "why not those?".

We would like to thank you for your warm words and professional remarks. Several changes were conducted within the manuscript according to your excellent suggestions (in “track changes”).

  1. Low level question: Besides cipro, what other interventions would the authors consider to be of interest? Are there any non-antibiotic adjuncts? 

We have previously demonstrated a positive trend with steroids, however it was not as effective as cipro. Increased survival and improved prognosis were also demonstrated with doxycycline (Gal et al. 2014). In addition, we have promising results with sulfisoxazole (we are planning to publish these results in the future). We use it as an endothelin receptor blocker (there is no meaning to an antibiotic effect, of course, but as a control we administered comparative doses of sulfamethoxazole/trimthoprim, which does not block endothelin, and this treatment was not efficient at all). To conclude, we believe that immunomodulators are excellent candidates, including IL-6 receptor antagonists, JAK inhibitors etc. (drugs that mitigated the “cytokine storm” in COVID-19 patients, for example).

Reviewer 2 Report

Comments and Suggestions for Authors

This paper discusses the long term damage associated with the treatment of ricin exposure by anti-toxin in a lung environment. 

The paper is reasonably basic in it approach to answer the question it poses at the start of the introduction (lines 63-64), but the overall premise of the paper is sound and of interest to the toxin community in my opinion. There are however some holes that I think could be filled to make the paper better. 

Specific area's where it would be great to see some improvements are as follows:

All histology figures could do with more information regarding the differences between the shame and treated imagery, for example highlighting area where there are key differences makes it easier for the reader to see without having to search especially those with less familiarity to histology. Scales bars could do with being bigger too.

It is a shame that the authors haven't assess more immunological readouts in this study, they speak of cytokine storms and neutrophil influx a number of times but do not assess if there is still any issue with this after 30 day post exposure. If there are samples or data available to this end I would encourage the authors to include this in the paper as it would certainly strengthen the paper considerably. 

Intranasal administration of anti-toxin is a rare way to deliver such a therapy I would be intrigued to know why this was chosen as the method (It is unlikely to be a delivery route in humans for example). Direct delivery of anti-toxin to the lung in itself could cause some immunological issues to the animals, there doesn't appear to the an anti-toxin only control in any of the data sets. This is also horse derived anti-sera, yes it will be less immunogenic as it is F(ab')2 but it could also exacerbate the immune system, perhaps somethign in the conclusions to address this would be good. 

All figure graphs need to be of higher resolution the text is very pixelated

Figures 3 & 4 both state an n=3 yet there are more dots on the graph that 3, so what is each dot representing - it can't be each mouse, therefore is it each image assessed? This needs to be clarified. 

The number of replicated for each experiment is vastly different:

Figure 1 - 8,7,12-16

Figure 2 - n=3

Figure 3 - n=3

Figure 4 - n=3

Figure 5 - n=5 (9 per group) - this is confusing is it 5 or 9 as there is 5 in one and 9 in the other group. 

Figure 6 - n = 28,31,12 

Why are the numbers so vastly different between each data type. I would envisage that some animal were used for some experiment and some for different ones, but it needs to be explained in the methods how the mice were divided for each experiment other wise it is quite confusing. 

Lines 195-198 - It is encouraging to see the authors state some limitations of their study, I would have likes to see an interim time point say 15 days in the data set too again this would have strengthened the paper. I understand it is high unlikely to be able to happen now but it is some thing to bare in mind for future work.

Author Response

Manuscript ID: toxins 2797472

Response to reviewer 2:

This paper discusses the long term damage associated with the treatment of ricin exposure by anti-toxin in a lung environment. 

The paper is reasonably basic in it approach to answer the question it poses at the start of the introduction (lines 63-64), but the overall premise of the paper is sound and of interest to the toxin community in my opinion. There are however some holes that I think could be filled to make the paper better. 

Specific area's where it would be great to see some improvements are as follows:

All histology figures could do with more information regarding the differences between the shame and treated imagery, for example highlighting area where there are key differences makes it easier for the reader to see without having to search especially those with less familiarity to histology. Scales bars could do with being bigger too.

According to the Reviewer suggestion, we highlighted the damaged areas in the images and described in more details the injury in the figure legend of Fig. 2 and 3. The scale bars in the images were enlarged and now are more visible.

It is a shame that the authors haven't assess more immunological readouts in this study, they speak of cytokine storms and neutrophil influx a number of times but do not assess if there is still any issue with this after 30 day post exposure. If there are samples or data available to this end I would encourage the authors to include this in the paper as it would certainly strengthen the paper considerably. 

Thank you for this excellent suggestion. We added the following sentence to the “Discussion section”: “It should be mentioned that this damage is vastly different from short-term damage (i.e. the damage recorded at 72 hours PE), since there are no neutrophils (Figure 2), pro-inflammatory cytokines, nor oxidative damage markers detected at 30 days PE (data not shown)”. We would like to mention that we have tested pro-inflammatory cytokines such as IL-6, TNFα and IL-10 and they were not detected at all at 30 PE. In addition, there was no difference between sham and ricin-intoxicated mice when comparing damage markers, i.e. xanthine oxidase (oxidative damage). Since these differences were not detected, we did not include the data in the result section, but only in the discussion. We will be more than happy to provide these results to the Reviewer upon request.

Intranasal administration of anti-toxin is a rare way to deliver such a therapy I would be intrigued to know why this was chosen as the method (It is unlikely to be a delivery route in humans for example).

We completely agree with this remark. This work, although published nowadays, was conducted several years ago. At that times, the antitoxin was administered intranasally. In current experiments, the antitoxin is administered intravenously. However, it is highly important to mention that treatment outcome is similar (please see more explanation in the next question).

Direct delivery of anti-toxin to the lung in itself could cause some immunological issues to the animals, there doesn't appear to the an anti-toxin only control in any of the data sets. This is also horse derived anti-sera, yes it will be less immunogenic as it is F(ab')2 but it could also exacerbate the immune system, perhaps somethign in the conclusions to address this would be good.

We previously demonstrated (Falach et al. 2018, please see reference # 8 in this manuscript) that survival ratios of i.n.- or i.v.- treated mice with the antitoxin are exactly the same. The whole idea in administrating F(ab')2 is to prevent serum-sickness. A long-term safety study that supports this notion was conducted  with other equine F(ab')2 antibodies produced here in IIBR (data not published). In addition, to the best of our knowledge, the adverse events related to antiotoxin administration are considered short-term effects (and are much more relevant if the antitoxin is administered parenterally (i.e. intravenously) in comparison to local administration.

All figure graphs need to be of higher resolution the text is very pixelated

We improved the graphs according to the reviewer suggestion.

Figures 3 & 4 both state an n=3 yet there are more dots on the graph that 3, so what is each dot representing - it can't be each mouse, therefore is it each image assessed? This needs to be clarified. 

Thank you for the comment. We clarified in this respect the figure legend of figures 3 and 4. The dots represent fields of view taken from 3 mice per group.

The number of replicated for each experiment is vastly different:

Figure 1 - 8,7,12-16

Figure 2 - n=3

Figure 3 - n=3

Figure 4 - n=3

Figure 5 - n=5 (9 per group) - this is confusing is it 5 or 9 as there is 5 in one and 9 in the other group. 

Figure 6 - n = 28,31,12 

Why are the numbers so vastly different between each data type. I would envisage that some animal were used for some experiment and some for different ones, but it needs to be explained in the methods how the mice were divided for each experiment other wise it is quite confusing.

We appreciate and thank the Reviewer for this highly important remark. Out of 12 surviving mice, 3 were taken to histology (H&E, Masson’s, immunohistochemistry), the rest (5 from the sham group and 9 of the surviving animals) were taken for lung-to-body weight determination. The compliance data (Figure 6) was collected from a separate experiment, however we added to this data results from other experiments only for “sham” and “antitoxin-treated” animals (results obtained prior to this experiment). In addition, we added to section 4.3 (animal experiments), the following sentence: “The determination of lung compliance (Figure 6) was conducted as a separate experiment”. 

Lines 195-198 - It is encouraging to see the authors state some limitations of their study, I would have likes to see an interim time point say 15 days in the data set too again this would have strengthened the paper. I understand it is high unlikely to be able to happen now but it is some thing to bare in mind for future work.

We thank the reviewer for the comment.  In previous studies we analyzed the damage at 14-15 days post exposure, for the determination of short-term damage. In this respect, all our experiments in intoxicated mice last two weeks, since survival may change until this time point. Therefore, in our experimental set we do not consider 15 days as long term period following intoxication. As you can see in the attached table (for the reviewer only), at the time point of 15 days post exposure we couldn’t detect pro-inflammatory cytokines or any clue of oxidative damages for assessment of inflammation and acute lung injury. Moreover, the parameters values tested 15 days post exposure are not statistical different from those tested at 30 days post exposure.

Tested Parameter     

Sham   (pg/ml)

Ricin              72h           (pg/ml)

Ricin + antitoxin day 15        (pg/ml)

Ricin + antitoxin day 30      (pg/ml)

TNF alpha

0 ± 0

16 ± 18*

3 ± 6

11 ± 15

IL-6

0 ± 0

3547 ± 1372*

0 ± 0

0 ± 0

XO

0.5 ± 0.1

4.1 ± 1.5*

0.974 ± 0.4

0.734 ± 0.3

* The only statistical changes in comparison to sham were at 72h